# Using Constrained Square-Root Cubature Kalman Filter for Quantifying the Severity of Epileptic Activities in Mice

**DOI:** 10.3390/biomedicines10071588

**Published:** 2022-07-03

**Authors:** Chih-Hsu Huang, Peng-Hsiang Wang, Ming-Shaung Ju, Chou-Ching K. Lin

**Affiliations:** 1Department of Neurology, National Cheng Kung University Hospital, College of Medicine, National Cheng Kung University, Tainan 70101, Taiwan; chihhsu66@gmail.com; 2Medical Device Innovation Center, National Cheng Kung University, Tainan 70101, Taiwan; 3Department of Mechanical Engineering, National Cheng Kung University, Tainan 70101, Taiwan; n16084983@gs.ncku.edu.tw

**Keywords:** constrained square-root cubature Kalman filter, temporal lobe epilepsy, severity of epileptiform discharge

## Abstract

(1) Background: Quantification of severity of epileptic activities, especially during electrical stimulation, is an unmet need for seizure control and evaluation of therapeutic efficacy. In this study, a parameter ratio derived from constrained square-root cubature Kalman filter (CSCKF) was formulated to quantify the excitability of local neural network and compared with three commonly used indicators, namely, band power, Teager energy operator, and sample entropy, to objectively determine their effectiveness in quantifying the severity of epileptiform discharges in mice. (2) Methods: A set of one normal and four types of epileptic EEGs was generated by a mathematical model. EEG data of epileptiform discharges during two types of electrical stimulation were recorded in 20 mice. Then, EEG segments of 5 s in length before, during and after the real and sham stimulation were collected. Both simulated and experimental data were used to compare the consistency and differences among the performance indicators. (3) Results: For the experimental data, the results of the four indicators were inconsistent during both types of electrical stimulation, although there was a trend that seizure severity changed with the indicators. For the simulated data, when the simulated EEG segments were used, the results of all four indicators were similar; however, this trend did not match the trend of excitability of the model network. In the model output which retained the DC component, except for the CSCKF parameter ratio, the results of the other three indicators were almost identical to those using the simulated EEG. For CSCKF, the parameter ratio faithfully reflected the excitability of the neural network. (4) Conclusion: For common EEG, CSCKF did not outperform other commonly used performance indicators. However, for EEG with a preserved DC component, CSCKF had the potential to quantify the excitability of the neural network and the associated severity of epileptiform discharges.

## 1. Introduction

Epilepsy is a common disorder of the central nervous system caused by abnormal or excessively synchronized firing of neurons. Sixty percent of adults with epilepsy are classified as having focal epilepsy, with temporal lobe epilepsy accounting for most cases [1,2]. Since the foci of temporal lobe epilepsy are commonly found in the hippocampus, many studies on epilepsy have focused on this region [3]. Oral antiepileptic drugs are currently the most common epilepsy treatment, however about 30% of patients cannot be effectively treated with drugs. Many alternative treatments have been investigated, including electrical stimulation [4,5,6,7]. Many parameters and waveforms can be adjusted when designing electrical stimulation protocols. The frequency range was roughly divided into low frequencies (<10 Hz) and high frequencies (greater than 50 Hz) [4]. Low frequency stimulation has been shown to reduce seizure frequency in vitro [8] and to suppress the kindled seizures of amygdala in vivo [9]. High frequency stimulation (HFS) also reduces seizure frequency. In in silico and in vitro conditions, it is believed that suppression of neuronal activity with HFS is generated by depolarization block [10,11,12,13]. Two in-vivo studies using the kindling epilepsy model showed that HFS in 130 Hz reduced tissue excitability and modified epileptogenesis of the tissue [14,15]. Regular rectangular pulse train stimulation (RPS) in HFS is the most frequently adopted waveform. In recent years, the therapeutic feasibility of random noise stimulation (RNS) have shown promise. The effects of RNS may depend on the amplitude of RNS. At a smaller amplitude, RNS may be facilitatory through stochastic resonance [16] or spike timing dependent synaptic plasticity [17]. On the contrary, at a larger amplitude, RNS may take over the neural circuit, exciting the neuron population randomly and interrupting the regularity of spikes. Preliminary clinical results have indicated that RNS can increase cortical excitability [18] and enhance the subjects’ working memory [19].

However, quantifying the severity of epileptiform discharge is an unmet need for seizure control and evaluation of therapeutic efficacy. The detection of spikes, the hallmark of epileptiform discharge, in EEG is a classical problem for which many indicators have been proposed [20], including those derived from time domain, frequency domain, time-frequency domain, and nonlinear methods. Recently, wavelet decomposition and empirical mode decomposition have also been proposed [21]. One study showed that using machine learning to analyze a set of indicators could increase the accuracy to nearly 100% [22,23]. However, most of the proposed algorithms are aimed at detecting the existence but not quantifying the severity of epileptiform discharge or excitability of the network [20]. Most of the studies intuitively used spike counts per unit time [24] or amplitude [25] as the quantifier without explicit evidence of its fidelity. Non-linear measures, such as entropy, were also used for quantifying the severity of epileptiform discharge implicitly, though their main function was seizure detection in some studies [26,27]. Recently, more studies have reported using machine learning and artificial intelligence for epileptic seizure detection [28,29]. Model-based methods are usually more sophisticated approaches. Kalman filter is probably the most widely used method to estimate state variables of linear models. However, commonly used seizure models are nonlinear and time-varying [30,31]. The Kalman filter has been modified in several ways, such as an extended or cubature Kalman filter (CKF) [32], to overcome the nonlinearity problem. In addition, triangular square-root factorization of the error covariance matrix has been shown to increase the stability of CKF. For time-varying systems, one proposed solution is to also treat the time-varying parameters of the model as state variables with a slower change rate [33].

The main purpose of this study was to investigate the eligibility of using constrained square-root cubature Kalman filter (CSCKF) to quantify the severity of epileptic activities. The performance of CSCKF was compared with three commonly used indicators, namely, band power, Teager energy operator, and sample entropy, using epileptic activities from both experiments and a computational seizure model.

## 2. Materials and Methods

### 2.1. Neural Mass Model of Mice Hippocampal Epilepsy

Figure 1 depicts a physiologically based neural mass model of electrical activities of hippocampus [34]. The model includes a pyramidal cell population, fast and slow inhibitory interneuron populations, and an excitatory interneuron population. The relationship between the presynaptic firing rate and postsynaptic membrane potential of each neuron population was modeled using a second-order linear transfer function. The postsynaptic membrane potential was then transformed into the firing rate of the postsynaptic neuron population using a sigmoid function (*S*),
(1)$$S(x)=\frac{2e_{s}}{1+e^{h_{s}(v_{s}-x)}}$$where *e_s_*, *h_s_*, and *v_s_* are constants, and their values are listed in Table A1 of the Appendix B. The corresponding state-space equations are given below.
(2)$$\begin{array}{l} {\dot{x}}_{1}=x_{6}\\ {\dot{x}}_{2}=x_{7}\\ {\dot{x}}_{3}=x_{8}\\ {\dot{x}}_{4}=x_{9}\\ {\dot{x}}_{5}=x_{10}\\ {\dot{x}}_{6}=Aa\;S(x_{2}-x_{3}-x_{4})-2ax_{6}-a^{2}x_{1}\\ {\dot{x}}_{7}=Aa\;(u_{1}+C_{2}\;S(C_{1}x_{1}))-2ax_{7}-a^{2}x_{2}\\ {\dot{x}}_{8}=Bb\;C_{4}\;S(C_{3}x_{1})-2bx_{8}-b^{2}x_{3}\\ {\dot{x}}_{9}=Gg\;\;C_{7}\;S(C_{5}x_{1}+C_{6}x_{5})-2gx_{9}-g^{2}x_{4}\\ {\dot{x}}_{10}=Bb\;S(C_{3}x_{1})-2bx_{10}-b^{2}x_{5}\\ {\dot{x}}_{11}=-0.5x_{11}+(x_{2}-x_{3}-x_{4})\\ y=C_{S}x_{11} \end{array}$$
where *C*_i_, i = 1 to 7, and *C_S_*, *a*, *b*, *g*, and *G* are constants, and their values are listed in Table A1 of Appendix B. *A* and *B* are parameters that are adjusted to generate normal and epileptic EEGs. The right-most transfer function, s/(s + 0.5), is a high pass filter which emulates the signal processing of an EEG recorder. The output, *y*, is the simulated EEG.

### 2.2. Formulation of CSCKF and Bifurcation Analysis

In this section, CSC−F was used to estimate the states of the mathematical seizure model, as shown in Figure 1, as well as the time-varying parameter B(t). First, the state space model (Equation (2)) was expanded,
(3)$$\begin{array}{l} {\dot{x}}_{c}=f_{c}(x_{c},t)+w_{c}(t)\\ y_{c}(t)=h_{c}(x_{c},t)+v_{c}(t) \end{array}$$
where **w_c_**(*t*) and *ν_c_*(*t*) are the state and output noise, respectively, in which Gaussian distribution was assumed, with zero mean and covariance matrix **Q** and **R**, respectively. **x***_c_* is the augmented state variable array,
(4)$$x_{c}=[x_{1}\;x_{2}\;\cdots \;x_{12}]$$
where *x*_1_ to *x*_11_ are the original state variables and *x*_12_ corresponds to B(t). In other words, this approach assumes that B(t) is a state variable. On the other hand, A(t) and G(t) are time-invariant constants, i.e., A(t) = 5 and G(t) = 10. In this way, *f_c_(**x_c_**,t)* and *h_c_(**x_c_**,t)* were redefined as
(5)$$\begin{array}{l} f_{c}(x_{c},t)=\left[\begin{array}{l} x_{6}\\ x_{7}\\ x_{8}\\ x_{9}\\ x_{10}\\ Aa\;S(x_{2}-x_{3}-x_{4})-2ax_{6}-a^{2}x_{1}\\ Aa\;(u_{1}+C_{2}\;S(C_{1}x_{1}))-2ax_{7}-a^{2}x_{2}\\ x_{12}b\;C_{4}\;S(C_{3}x_{1})-2bx_{8}-b^{2}x_{3}\\ Gg\;\;C_{7}\;S(C_{5}x_{1}+C_{6}x_{5})-2gx_{9}-g^{2}x_{4}\\ x_{12}b\;S(C_{3}x_{1})-2bx_{10}-b^{2}x_{5}\\ -0.5x_{11}+(x_{7}-x_{8}-x_{9})\\ \begin{array}{l} 0 \end{array} \end{array}\right] \end{array}$$
(6)$$h_{c}(x_{c},t)=c_{s}x_{11}$$

Because the experimental data were all digitalized signals, the next step was to digitize the model, i.e.,
(7)$$\begin{array}{l} {\dot{x}}_{c}(t(k))≃\frac{x_{k}-x_{k+1}}{T}=f_{c}(x_{k-1},u_{k-1})+w_{c}(t(k-1))\\ \Rightarrow \;x_{k}=x_{k-1}+T\cdot (f_{c}(x_{k-1},u_{k-1})+w_{c}(t(k-1)))\\ \;\;\;=f(x_{k-1},u_{k-1})+w_{k-1}\\ y_{k}=y_{c}(t(k))=h_{c}(x_{c},t)+v_{k}=c_{s}x_{11}+v_{k} \end{array}$$
where the sampling rate is *1/T*, *k* is the indexed time point, and **x*_c_****(t(k))* and *y_c_(t(k))* are abbreviated as **x***_k_* and *y_k_*, respectively.

The Kalman filter was calculated using a recursive algorithm of two alternating steps. Step 1 was the time update, i.e., updating the state estimates to the present time (k) based on the data of previous time (*k* − 1). **S***_k_*_−1|*k*−1_ was obtained from step 2 of the previous time point (*k* − 1). The states at the cubature points (*i* = 1, 2 … *m*; *m* = 2n_x_) of the previous and present times were estimated,
(8)$$\begin{array}{l} X_{i,k-1|k-1}=S_{k-1|k-1}\xi_{i}+{\hat{x}}_{k-1|k-1}\\ X_{i,k|k-1}^{*}=f(X_{i,k-1|k-1},u_{k-1}) \end{array}$$
where [*ζ_i_*] represents the cubature points. Using the definition of cubature points, the estimates of present states based on the data up to the previous time point were obtained,
(9)$${\hat{x}}_{k|k-1}=\frac{1}{m}\sum_{i=1}^{m}X_{i,k|k-1}^{*}$$
and the square-root factor (**S***_k|k_*_−1_) of the predicted error covariance was updated to the present time,
(10)$$S_{k|k-1}=\mathrm{Tria}\left(\left[\chi_{k|k-1}^{*}\;S_{Q,k-1}\right]\right)$$
where Tria is a general triangularization function to produce a lower triangular matrix, and
(11)$$\begin{array}{l} \chi_{k|k-1}^{*}=\frac{1}{\sqrt{m}}\left[X_{1,k|k-1}^{*}-{\hat{x}}_{k|k-1}\;\;X_{2,k|k-1}^{*}-{\hat{x}}_{k|k-1}\;\;\ldots \;\;X_{m,k|k-1}^{*}-{\hat{x}}_{k|k-1}\;\right]\\ S_{Q,k-1}S_{Q,k-1}^{T}=Q_{k-1} \end{array}$$▌ (End of step 1)

Step 2 was the measurement update, i.e., updating the data to be used when estimating the present states to the present time. First, **S***_k_*_|*k*−1_ was obtained from the results of step 1, and the states (X*_i,k|k_*_−1_) and system output (**y***_i,k|k_*_−1_) at the cubature points (*i* = 1, 2 … m; m = 2n_x_) of the present time were estimated,
(12)$$\begin{array}{l} X_{i,k|k-1}=S_{k|k-1}\xi_{i}+{\hat{x}}_{k|k-1}\\ y_{i,k|k-1}=h(X_{i,k|k-1}) \end{array}$$

Then, using the definition of cubature points, the estimate of system output at the present time point was obtained,
(13)$${\hat{y}}_{k|k-1}=\frac{1}{m}\sum_{i=1}^{m}y_{i,k|k-1}$$
and the square-root factor of the predicted error covariance was updated to the present time,
(14)$$S_{yy,k|k-1}=\mathrm{Tria}\left(\left[Y_{k|k-1}\;\;S_{R,k}\right]\right)$$
where
(15)$$\begin{array}{l} Y_{k|k-1}=\frac{1}{\sqrt{m}}\left[Y_{1,k|k-1}-{\hat{x}}_{k|k-1}\;\;Y_{2,k|k-1}-{\hat{x}}_{k|k-1}\;\;\ldots \;\;Y_{m,k|k-1}-{\hat{x}}_{k|k-1}\right]\\ S_{R,k}S_{R,k}^{T}=R_{k} \end{array}$$

Next, cross-covariance (*P_xy,k|k_*_−1_) was calculated,
(16)$$\begin{array}{l} P_{xy,k|k-1}=\chi_{k|k-1}Y_{k|k-1}^{T}\\ \chi_{k|k-1}=\frac{1}{\sqrt{m}}\left[X_{1,k|k-1}-{\hat{x}}_{k|k-1}\;\;X_{2,k|k-1}-{\hat{x}}_{k|k-1}\;\;\ldots \;\;X_{m,k|k-1}-{\hat{x}}_{k|k-1}\;\right] \end{array}$$
and Kalman gain (*W_k_*) and states were updated to the current time point (*k*) based on the data of the present time,
(17)$$\begin{array}{l} W_{k}=(P_{xy,k|k-1}/S_{yy,k|k-1}^{T})/S_{yy,k|k-1}\\ {\hat{x}}_{k|k}={\hat{x}}_{k|k-1}+W_{k}(y_{k}-{\hat{y}}_{k|k-1}) \end{array}$$

Finally, the square-root factor (**S***_k|k_*) of the predicted error covariance was updated based on the data of the present time,
(18)$$S_{k|k}=\mathrm{Tria}\left(\left[\chi_{k|k-1}-W_{k}Y_{k|k-1}\;\;W_{k}S_{R,k}\right]\right)$$▌ (End of step 2)

The ratio I_B/A_ = *x_12_*/A, representing the estimated B(t)/A(t), was used as the indicator of excitability of the model in the later analyses. Potential types of dynamic behavior and bifurcation in the epilepsy model were analyzed using MatCont software [35].

### 2.3. Animal Preparation and Experiment Setup

The C57BL/6 mice were housed under standard conditions in the animal facility of the College of Medicine, National Cheng Kung University (NCKU) and kept on a 12 h light/dark cycle with food and water ad libitum. The study was conducted according to the guidelines of the Declaration of Helsinki, and approved by the Institutional Review Board of National Cheng Kung University (IACUC Approval No: 108223).

A set of three electrodes with two electrodes in spiral form were implanted into the CA3 region of the hippocampus (AP − 1.34 mm, ML ± 1.25 mm and DV − 1.8 mm from the bregma) using a stereotaxic technique under isoflurane (Attane, Step, Taipei, Taiwan) anesthesia. Two stainless steel screws were fixed on the skull above the cerebellum and olfactory bulb as the ground and reference electrodes, respectively. The spiral electrodes were used for electrical stimulation and the other implanted electrode and the screw electrodes were used for EEG recording. The experiments were performed 2 weeks after electrode insertion when the animals had fully recovered from surgery.

Acute seizures were induced by intraperitoneal (ip) injection of a bolus of kainic acid (ab120100, abcam, Boston, MA, USA) (10 mg/kg), followed by boluses of 5 mg/kg per 0.5 h until the behavior of mice was rated as 5 on the Racine scale. The injection rate was then reduced to boluses of 2.5 mg/kg/h. After the experiment, diazepam (Dupin, China Chemical & Pharmaceutical Co., Taipei, Taiwan) (ip: 10 mg/kg) was given to stop the seizures.

EEG signals were first amplified 1000× and hardware filtered (0.1 Hz–1000 Hz bandpass and 60 Hz notch filters) by a biopotential amplifier (Differential AC Amplifier model 1700, A-M System, Sequim, WA, USA), and then sampled at 3840 Hz into a personal computer using a control system development tool (DS1104, dSPACE, Paderborn, Germany). The behavior of the mice was videotaped throughout the experiment to assist later analysis.

The DS1104 also output control signals to drive an electrical stimulator (A395 Linear Stimulus Isolator, World Precision Instruments, Sarasota, FL, USA) to produce electrical stimulation in constant current mode. For the RNS, the control signal was band-limited (101–640 Hz) white noise, and the amplitude was adjusted so that the extreme amplitude of actual current output was ±500 μA. For the RPS, the control signal was a sequence (5 s) of 128 Hz biphasic and charge balanced square-wave pulses, with the positive phase 458.5 μA in height and 781.25 μs in duration, and the negative phase −50.94 μA in height and 7031.25 μs in duration. Sham stimulation had identical waveform parameters as the real stimulation, except the height was only 1% that of the real stimulation.

The electrical stimulation was given manually when type III epileptic waveforms, i.e., 4–12 Hz spikes lasting longer than 5 s, were detected by visual inspection. The rationale for choosing this type of seizure was (1) that the animal had a Racine score of 3 and showed no obvious convulsive movements, thus preventing injury to the animal and electrode setup, and (2) that it was relatively objective to judge whether epileptic activities were suppressed.

In total, 20 C57BL/6 mice (age: 46.2 ± 5.6 weeks and body weight: 33.3 ± 4.5 g) were used in this study, of which 6 received RNS, 9 received RPS, and the remaining 5 were the controls and did not receive electrical stimulation. The stimulation shifted between real and sham stimulation pseudo-randomly. Consecutive stimulations were separated by at least 1 min, and repeated until no more consistent type III epileptic waveforms were noted.

EEG was recorded throughout the whole experiment. A 209-order equiripple finite impulse response low-pass filter with a pass band below 50 Hz, a stop band above 100 Hz, pass band ripple less than 0.1 dB, and stop band attenuation of 60 dB was used to remove stimulation artifacts.

### 2.4. Formulation of Other Quantifying Indicators of Epileptic Activities

Another three indicators were formulated for comparison. The first was band power (P_b_) in the frequency range of 4–50 Hz. The second was Teager energy operator (*E_T_*) [36].
(19)$$E_{T}=log_{10}\left(\frac{1}{N}\;\sum_{k=4}^{N+3}\left(x[k-1]\cdot x[k-2]-x[k]\cdot x[k-3]\right)\right)$$

The third was sample entropy (*SEn*) [26,37], a variant of approximate entropy with two advantages: data length independence and relatively trouble-free implementation. Sample entropy was calculated as follows [26]. Initially, a set of vectors, **X***_m_*(*i*), was formed from an EEG segment of length *N*, **X***_m_*(*i*) = [*x*(*i*), *x*(*i* + 1), … *x*(*i* + *m* − 1)], where *m* is a constant integer. The distance function, d[X*_m_*(*i*), X*_m_*(*j*)], was defined as the Chebyshev distance and the number of vector pairs of d[X*_m_*(*i*), X*_m_*(*j*)] < *r*∙SD and d[X*_m_*_+1_(*i*), X*_m_*_+1_(*j*)] < *r*∙SD, where *r* is a constant and SD is the standard deviation of the EEG segment about the mean, denoted as B*_m_*(*r*) and A*_m_*_+1_(*r*), respectively. *SEn* was then defined as
(20)$$SEn(m,r,N)=ln\left(\frac{B_{m}(r)}{A_{m+1}(r)}\right)$$

In this study, *m*, *r*, and *N* were set as 4, 0.1, and 2400, respectively. The time-window width of the data segment to calculate all three indicators was 5 s.

All the calculations and simulations were carried out using the commercial software package, Matlab and Simulink (MathWorks, Natick, MA, USA).

### 2.5. Statistical Analyses

Group results were summarized as mean ± one standard deviation. Two-way mixed design ANOVA with treatment, i.e., electrical stimulation and time (before, during and after stimulation) as the main factors, and post-hoc analysis by non-parametric Wilcoxon signed-rank test were used to test the efficacy of electrical stimulation. The significance level was set at *p* < 0.05. Multiple comparisons using non-parametric Mann–Whitney U tests with Bonferroni correction were used to test the difference of performance indicator between types of EEG. The significance level was set at α < 0.01.

## 3. Results

### 3.1. Performance of Quantitative Indicators in Simulated EEG

The mathematical model described in the Methods section was used to generate three sets of one normal and four types of epileptic activities as shown in Figure A1 in Appendix A. Each combination of A and B was used to generate 50 EEG segments of 5 s. The excitability of the neural network model decreased from Type I to Type IV, i.e., the severity of epileptic discharges decreased from Type I to Type IV.

The results of the four indicators are shown in Figure 2. The results of I_B/A_ largely deviated from those expected, i.e., I_B/A_ would increase from Type I (around 3) to Normal (around 10). For Type I epileptiform discharge, some estimated I_B/A_ values were around 3 as expected, but many values were at the other extreme (around 15). The estimates of Types II, III, and IV, in the range of 1 to 4, were lower than those of Type I. The estimates of normal EEGs matched the expected values.

P_b_ showed a trend compatible with the subjective judgement, i.e., the power increased from Type I to Type II, and then decreased from Type II to Type IV. In addition, the corresponding P_b_ values also increased with A, the parameter of the model that made a major contribution to the amplitude of the spikes. The wave amplitudes of all types increased with A but at different rates, so that the relative amplitudes of Type I and Type IV could reverse due to different A values. The results of E_T_ and SEn were similar to those of P_b_. In short, all four indicators showed a trend compatible with subjective human judgement. However, the trend did not match the trend of excitability of the neural network, i.e., B/A.

Of note, the trends of the four indicators were quite similar. Except for Type I EEG, the severity of seizures decreased from Type 2 EEGs to normal EEGs. In addition, the variance was the largest for Type I EEGs in all four performance indicators, and especially for I_B/A_.

### 3.2. Performance of Quantitative Indicators in Simulated Output of the Seizure Model

The right-most transfer function after the model output, z(t) (Figure A2), was used to emulate the high-pass filtering properties of commonly used EEG machines (Figure 1). The split phenomenon in estimating parameters of Type 1 EEGs by CSCKF was suspected to be due to the filter. Therefore, the performance of the four quantitative indicators was re-investigated using z(t) as the output from the model (Figure 3). Except for I_B/A_ and P_b_, the results of the other three indicators were almost identical to those using y(t) as the output. For P_b_, all values increased, but the relative trend was maintained. For I_B/A_, the split phenomenon disappeared, and all estimates were concentrated around the expected value of 3 (Figure 4 right). In addition, the trend of estimates was consistent with the setting, i.e., graded increase of I_B/A_ from Type 1 EEGs to normal EEGs. The variance of estimates for each type of EEG also decreased.

### 3.3. Performance of I_A/B_ in the Experimental Data without Electrical Stimulation

The estimated I_B/A_ values of experimental EEG data without electrical stimulation are shown in Figure 5. It is clear that most of the normal EEGs were estimated to have a larger I_B/A_ value of around 10, while most of the epileptiform EEGs were estimated to have a lower I_B/A_ value of around 3. However, some epileptiform EEGs had higher I_B/A_ values than those of normal EEGs. In addition, the normal and epileptiform EEGs were segregated into two groups, and there were only a small number of intermediate points between these two peaks, indicating a trend of dichotomy. In other words, I_B/A_ could distinguish between normal and epileptiform discharges, but could not quantify their severity. This phenomenon, i.e., split of estimates to two ends in Type 1 EEGs, was also similar to those seen in simulated EEGs (Figure 2 and Figure 4).

### 3.4. Performance of Four Quantitative Indicators in RNS Experiments

Thirty triplets, before, during, and after stimulation, of EEG segments (5 s) from six mice receiving RNS were used in the following analysis (Figure 6). The group results (mean ± standard deviation) of I_B/A_ before, during, and after real stimulation were 5.80 ± 4.46, 5.80 ± 4.63, and 2.04 ± 0.92, respectively. Two-way mixed design ANOVA showed that both time (*F*_2,96_ = 10.80, *p* < 0.001) and treatment (*F*_1,48_ = 13.84, *p* < 0.001) were significant factors, and that there was a significant interaction between the effects of the two factors (*F*_2,96_ = 4.34, *p* = 0.016).

The group results (mean ± standard deviation) of P_b_ before, during, and after real stimulation were −3.02 ± 5.09, −4.78 ± 4.79, and −2.52 ± 4.39, respectively. Two-way mixed design ANOVA showed that time was a significant factor (*F*_2,116_ = 5.488, *p* = 0.005), but treatment was not (*F*_1,58_ = 0.043, *p* = 0.837), and there was a significant interaction between the effects of the two factors (*F*_2,116_ = 4.523, *p* = 0.013). The group results of E_T_ before, during, and after real stimulation were −2.99 ± 0.53, −2.76 ± 0.21, and −2.61 ± 0.24, respectively. Two-way mixed design ANOVA revealed that time was a significant factor (*F*_2,106_ = 10.581, *p* < 0.0001), but treatment was not (*F*_1,53_ = 2.83, *p* = 0.098), and there was a significant interaction between the effects of the two factors (*F*_2,106_ = 9.482, *p* < 0.001). The group results of SEn before, during, and after real stimulation were 0.33 ± 0.12, 0.62 ± 0.21, and 0.23 ± 0.12, respectively. Two-way mixed design ANOVA showed that both time (*F*_2,112_ = 49.51, *p* < 0.001) and treatment (*F*_1,56_ = 6.12, *p* = 0.016) were significant factors, and that there was a significant interaction between the effects of the two factors (*F*_2,112_ = 56.55, *p* < 0.001).

### 3.5. Performance of the Four Quantitative Indicators in RPS Experiments

Twenty-seven triplets before, during, and after stimulation of EEG segments (5 s) from nine mice receiving RPS were used in the following analysis (Figure 7). The group results (mean ± standard deviation) of I_B/A_ before, during, and after real stimulation were 1.84 ± 0.58, 5.95 ± 4.29, and 3.82 ± 4.81, respectively. Two-way mixed design ANOVA showed that neither time (*F*_2,96_ = 1.839, *p* = 0.165) nor treatment (*F*_1,48_ = 3.220, *p* =0.079) was a significant factor, but that there was a significant interaction between the effects of the two factors (*F*_2,96_ = 6.312, *p* = 0.003).

The group results (mean ± standard deviation) of P_b_ before, during, and after real stimulation were −0.53 ± 5.59, −5.72 ± 4.39, and −4.59 ± 7.54, respectively. Two-way mixed design ANOVA showed that time was a significant factor (*F*_2,102_ = 10.85, *p* < 0.001), but treatment was not (*F*_1,51_ = 0.12, *p* = 0.75), and there was a significant interaction between the effects of the two factors (*F*_2,102_ = 7.73, *p* < 0.001). The group results of E_T_ before, during, and after real stimulation were −2.68 ± 0.54, −3.27 ± 0.36, and −3.07 ± 0.62, respectively. Two-way mixed design ANOVA showed that time was a significant factor (*F*_1,52_ = 21.57, *p* < 0.001), while treatment was not (*F*_1,52_ = 1.677, *p* = 0.201), and there was a significant interaction between the effects of the two factors (*F*_2,104_ = 14.66, *p* < 0.001). The group results of Sen before, during, and after real stimulation were 0.29 ± 0.69, 0.31 ± 0.07, and 0.18 ± 0.08, respectively. Two-way mixed design ANOVA showed that time was a significant factor (*F*_2,92_ = 17.90, *p* < 0.001), while treatment was not (*F*_1,46_ = 0.331, *p* = 0.568), and there was a significant interaction between the effects of the two factors (*F*_2,92_ = 5.730, *p* = 0.005).

In summary, the results of the four indicators were inconsistent during both types of electrical stimulation. Because of the lack of a gold standard, it was unclear which indicator had the best performance.

### 3.6. Bifurcation Analysis

Because the results of the performance indicators were inconsistent, we investigated this issue using simulated data. Three sets of one normal and four types of epileptic activities were generated using a seizure model to evaluate the performance indicators, and the results of bifurcation analysis offered an explanation for the discrepancy in indicator performance when using different outputs (Figure 8, right plot). Within the range between the saddle-node (SN, at B/A = 7.824) and saddle-saddle (SS, at B/A = 12.897) bifurcation points, the dynamics of the epilepsy model had one stable and two unstable fixed points, while it had only either one stable or unstable fixed point outside this range. The dynamics underwent supercritical Andronov–Hopf (AH) bifurcation (negative Lyapunov number) at B/A = 2.478 as I_B/A_ increased from below, and then a limit cycle persisted until B/A reached the SN bifurcation point. As shown, the Type II epileptiform activity generated by the epilepsy model corresponded to the dynamics of limit cycles with small amplitudes, and the Type III epileptiform activity was generated by large-amplitude limit cycle dynamics. The irregular Type IV and clustered Type V epileptiform activities appeared when B/A was slightly larger than the SN bifurcation point. The irregular or clustered large spikes in Type IV and Type V epileptiform activities originated from the limit cycle behavior due to escape from the attraction domain of the stable fixed point (lower solid line) in the presence of noise in the model input to the unstable fixed point region (upper dashed lines). Normal activity was generated when B/A was sufficiently far away from the SN bifurcation point, so that the input noise was hardly able to lift z(t) to the unstable region.

The right plot of Figure 8 shows the results when the model output z(t) was used for analysis, and the left plot shows the results when the simulated EEG y(t) was used for analysis. Because of the high-pass filtering property of EEG machines, the DC component was removed from z(t), after which the stable points were all zero. As explained above, Type I EEGs were generated by the smallest B/A ratio, i.e., highest excitability, in four types of epileptiform discharge. The oscillation of Type I EEGs was very fast and small, and may have been obscured by background noise. If the DC component was removed by the high-pass filter of common EEG machines, it became difficult to differentiate the Type I EEGs from normal EEGs, as seen in the left plot of Figure 8. This is a problem of observability.

## 4. Discussion

In this study, the performance indicator derived from CSCKF was compared with three conventional performance indicators to evaluate the severity of epileptiform discharges using both simulated and experimental data. For the experimental data, the results of the four indicators were inconsistent during both types of electrical stimulation, although a trend of lower epileptiform discharges during electrical stimulation was observed. Due to the lack of a gold standard, it is unclear which indicator had the best performance. When the simulated EEG segments were used, the results of all four indicators were similar; however, this trend did not match the trend of the parameters of the model. When the model output was used, except for I_B/A_, the results of the other three indicators were almost identical. For I_B/A_, the estimates faithfully reflected the severity of epileptiform discharges.

The detection and quantification of seizure severity using electrophysiological measures is an important topic, and many signal processing techniques have been developed [38]. The main difficulty is the non-linearity and non-stationarity of epileptiform discharges. A simple and robust method to overcome the non-linearity and non-stationarity is therefore needed. Another issue is the lack of a gold standard. Conventionally, the severity of epileptiform discharge is assessed by experts through visual inspection. Most of the studies intuitively used spike counts per unit time (frequency equivalent) [24] or amplitude [25] as the quantifier without explicit evidence of its fidelity. When frequency is relatively stable, amplitude is proportional to band power (P_b_). Teager energy operator (E_T_) takes both frequency and amplitude into consideration. Entropy (SEn) is a nonlinear indicator that measures randomness, based on the fact that regularity increases during seizure attack [39]. However, there may not be consensus among experts, when frequency and amplitude change in opposite direction, as seen with the Type I and Type II seizure activities in Figure A1. From the formulae, P_b_ is the band power of a signal between 4 and 50 Hz. The more EEG spikes are present and the greater the amplitude of each spike, the higher the P_b_ value. However, if the frequency of the EEG spike is out of the defined range of P_b_, then this description may not be true. E_T_ is another method of calculating signal energy, and it can be used to quantify instantaneous changes in signal amplitude and frequency. The indicator E_T_ used in this study is the average value of E_T_ over a period of time, and it was also positively correlated with the amplitude and frequency of the EEG spikes. Sample entropy is a nonlinear metric used to quantify the regularity or complexity of a signal. In general, fewer variations in the spike frequency and amplitude will lead to lower sample entropy of the EEG signal. In addition, for EEG signals with similar spike frequencies, the general amplitude of the waveform has little effect on their sample entropy values. As seen in Figure A1, although there were clear differences in the general amplitudes between the three Type II signals, their SEn values were nearly identical. Therefore, special care should be taken when using sample entropy as an indicator of epileptiform discharge severity. In this paper we used the ratio of inhibitory weight to excitatory weight in the local neural network as the indicator, which is used as an indicator in a study focused on postictal generalized EEG suppression [40]. The advantage is that it is more theoretically based, while the disadvantage is that it is difficult to estimate.

RNS suppressed epileptiform discharges only during stimulation, while the suppressive effects of RPS persisted after the stimulation was stopped, as reveal by P_B_ and E_T_. E_T_ showed largest difference between RNS and RPS. However, if the post stimulation suppression was used as the indicator, RNS was superior to RPS. We think the contradiction may reflect the different suppressive mechanisms underlying these two types of stimulation. As is revealed by Figure 9, the amplitude of electrical activity decreased while the mean frequency unchanged, implying RNS partially reduced synchronization but the seizure loop persisted. On the other hand, RPS directly disrupted seizure loop and the main spike frequency disappeared. RPS possibly inhibits a spike train by maintaining a post-spike refractory period. The mechanism of RNS is still under active research. As mentioned in the Introduction, the effects of RNS may depend on the amplitude of RNS. At a larger amplitude, RNS may take over the neural circuit, exciting the neuron population randomly and interrupting the regularity of spikes.

The results of bifurcation analysis imply that it may be feasible to estimate the excitability of a neural network and quantify the severity of epileptiform discharges if DC EEG machines are used to record neural activities. However, the problem of more artefacts when using DC EEG machines has to be overcome. Further studies are needed to investigate this issue and validate our findings.

## 5. Conclusions

Based on the artificially generated and experimentally derived EEG data, the four performance indicators showed similar but varying trends, partially conforming to the severity of seizure activity. However, when the DC component of EEG was preserved as in the simulated data, only CSCKF could quantify the severity of seizure activity.

## Figures and Tables

**Figure 1 biomedicines-10-01588-f001:**
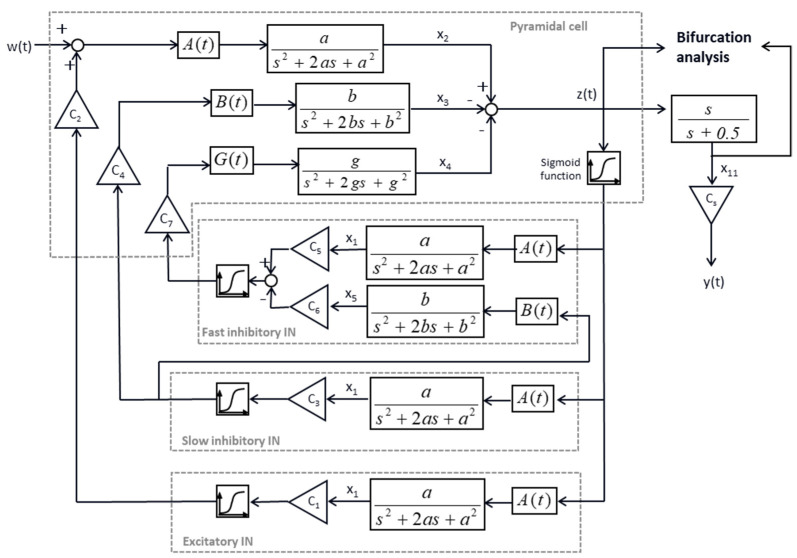
A physiologically based neural mass model of electrical activities of the hippocampus, adapted from [34].

**Figure 2 biomedicines-10-01588-f002:**
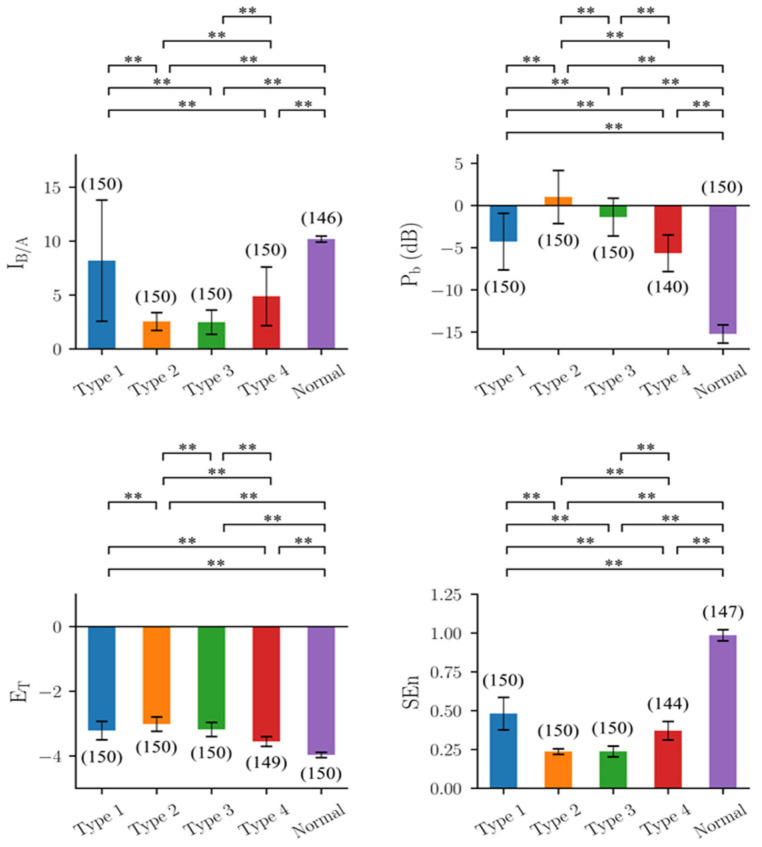
Results of the four indicators on the EEG segments produced by the mathematical seizure model, y(t). I_B/A_: estimated B/A from CSCKF, P_b_: band power in the frequency range of 4–50 Hz, E_T_: Teager energy operator, and SEn: sample entropy. The colored bar shows the mean value for each group, and the error bar shows the corresponding range of ±1 standard deviation. The number within the parentheses is the number of EEG segments, where the outliers have been excluded through the 1.5 inter-quartile range rule. **: Significant difference (*p*-value < 0.01) tested by multiple comparisons using non-parametric Mann–Whitney U tests with Bonferroni corrections with α = 0.01.

**Figure 3 biomedicines-10-01588-f003:**
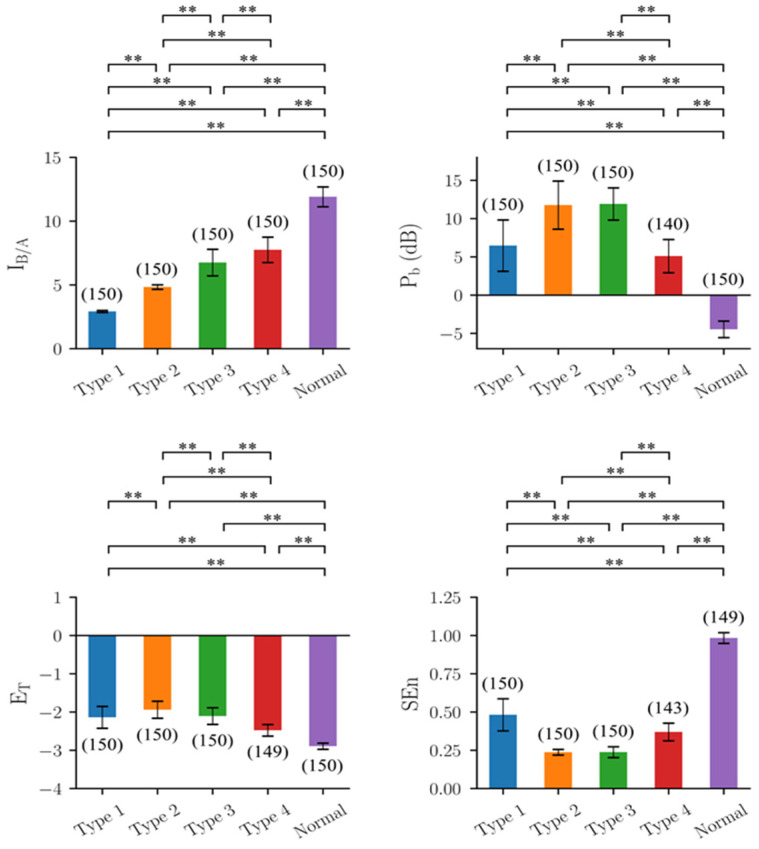
Results of the four indicators on the model output z(t) produced by the mathematical seizure model. I_B/A_: estimated B/A from CSCKF, P_b_: band power in the frequency range of 4–50 Hz, E_T_: Teager energy operator, and SEn: sample entropy. The colored bar shows the mean value for each group, and the error bar shows the corresponding range of ±1 standard deviation. The number within the parentheses is the number of EEG segments, where the outliers have been excluded through the 1.5 inter-quartile range rule. **: Significant difference (*p*-value < 0.01) tested by multiple comparisons using non-parametric Mann–Whitney U tests with Bonferroni corrections with α = 0.01.

**Figure 4 biomedicines-10-01588-f004:**
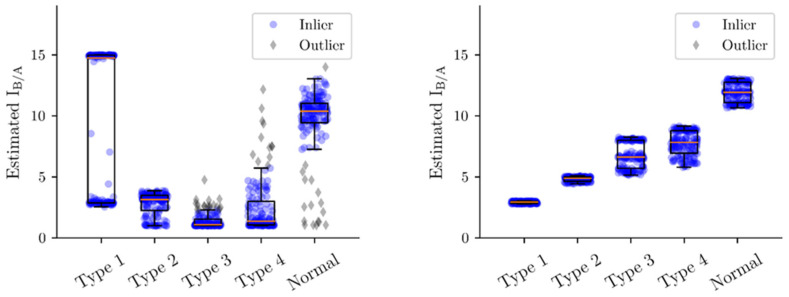
Scatter plots of I_B/A_ for y(t) **(left plot**) and z(t) (**right plot**), corresponding to the left upper plots of Figure 2 and Figure 3, respectively. In the case of z(t), the trend of estimates was consistent with the setting, i.e., graded increase of I_B/A_ from Type 1 EEGs to normal EEGs. In addition, three bands, corresponding to three values of A (Figure A1), can be seen. The boxplot for each group has a standard form, displaying the inter-quartile range (box), the range of inliers (error bar), and the median (orange line).

**Figure 5 biomedicines-10-01588-f005:**
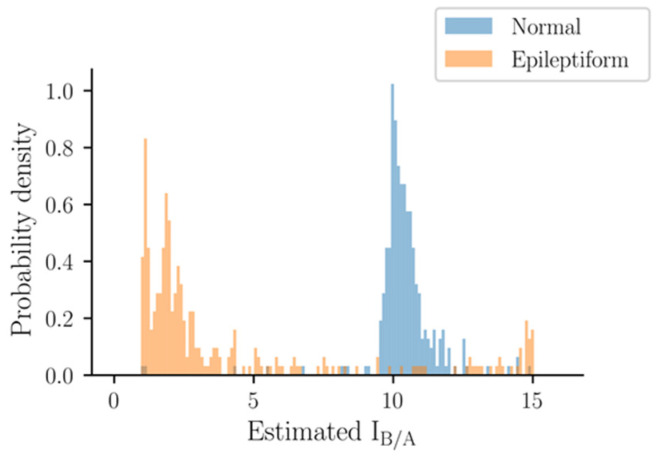
Estimated I_B/A_ values of the experimental EEG data without electrical stimulation.

**Figure 6 biomedicines-10-01588-f006:**
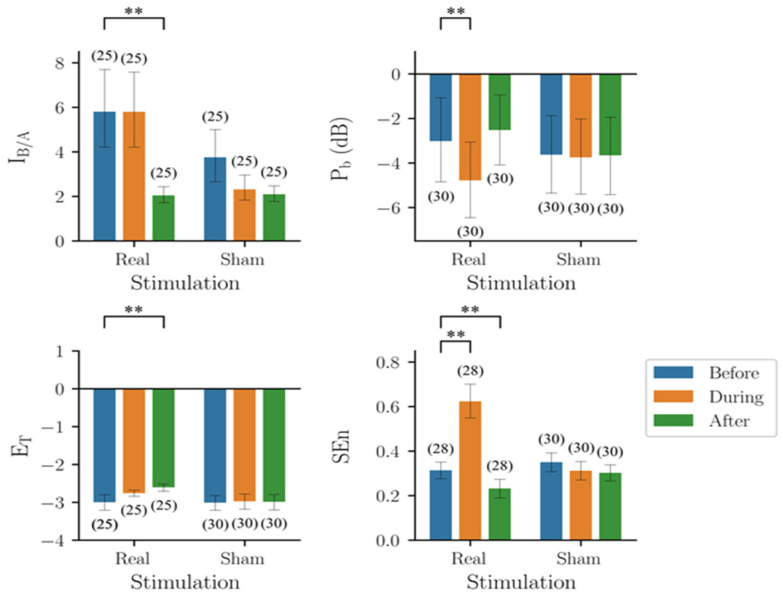
Results of the four indicators on EEG segments collected from mice before, during, and after random noise stimulation. I_B/A_: estimated B/A from CSCKF, P_b_: band power in the frequency range of 4–50 Hz, E_T_: Teager energy operator, and SEn: sample entropy. The colored bar shows the mean value of each group, and the error bar shows the corresponding range of ±1 standard deviation. The number within the parentheses is the number of EEG segments, where the outliers have been excluded by the 1.5 inter-quartile range rule. **: Significant difference (*p*-value < 0.01) was tested by using non-parametric Wilcoxon signed-rank tests.

**Figure 7 biomedicines-10-01588-f007:**
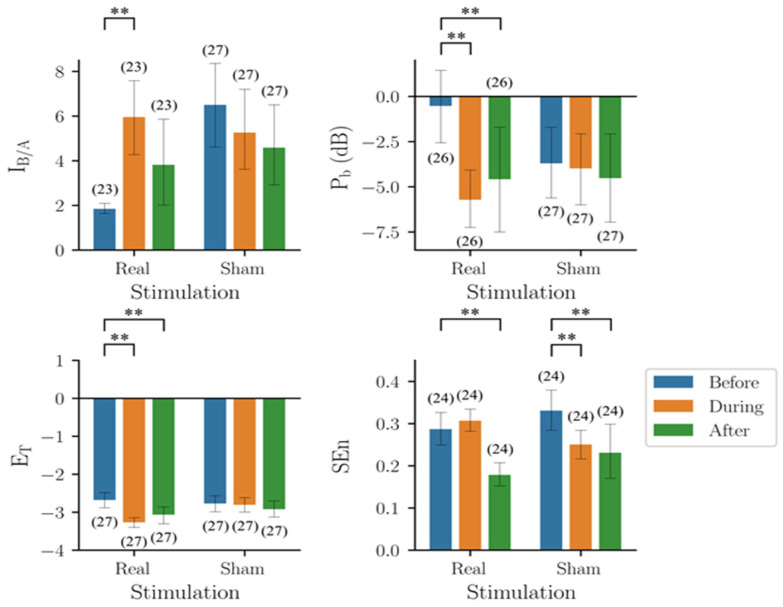
Results of the four indicators on the EEG segments collected from mice before, during, and after regular pulse stimulation. I_B/A_: estimated B/A from CSCKF, P_b_: band power in the frequency range of 4–50 Hz, E_T_: Teager energy operator, and SEn: sample entropy. The colored bar shows the mean value of each group, and the error bar shows the corresponding range of ±1 standard deviation. The number within the parentheses is the number of EEG segments, where the outliers have been excluded by the 1.5 inter-quartile range rule. **: Significant difference (*p*-value < 0.01) tested by using non-parametric Wilcoxon signed-rank tests.

**Figure 8 biomedicines-10-01588-f008:**
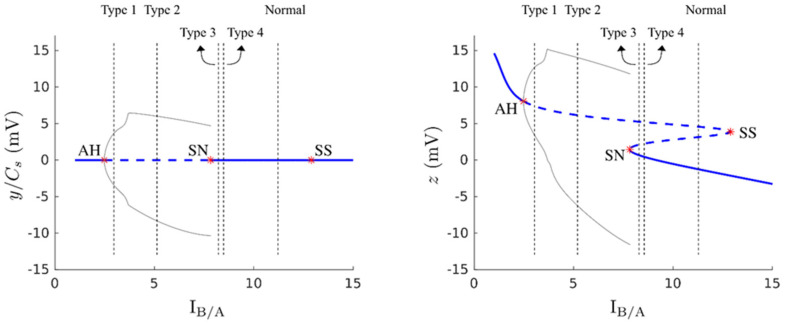
Bifurcation analysis of the model output, z, and the simulated EEG machine output, y, of the epilepsy model with respect to B/A. The blue line is the equilibrium continuation curve, where the solid segments correspond to stable fixed points and the dashed segment to unstable fixed points. Bifurcation points are represented by red stars. AH indicates the Andronov–Hopf bifurcation, SS the saddle–saddle bifurcation, and SN the saddle–node bifurcation. Gray solid curves correspond to the extremal values of limit cycles. The five vertical dashed black lines show the corresponding values of B/A used to generate the five types of simulated EEG activities when the parameter A was fixed at 5 (as shown in the upper row of Figure A1).

**Figure 9 biomedicines-10-01588-f009:**
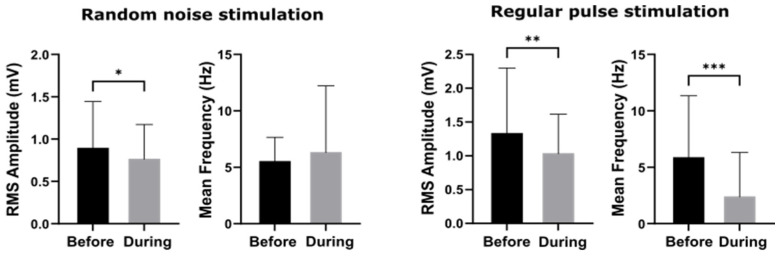
The group amplitude and mean frequency of EEGs before and during the two stimulation protocols, respectively. Significant difference (*p*-value *: < 0.05, **: < 0.01 and ***: < 0.001) was tested by using non-parametric Wilcoxon signed-rank tests.

## Data Availability

The data presented in this study are available on request from the corresponding author. The data are not publicly available due to its size.

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
