# Peer review of "Using Constrained Square-Root Cubature Kalman Filter for Quantifying the Severity of Epileptic Activities in Mice"

_biomedicines, 2022, doi:10.3390/biomedicines10071588_

Round 1

Reviewer 1 Report

Paper titled (Using constrained square-root cubature Kalman filter for quantifying the severity of epileptic activities in mice) by Huang et al. studied the utility of  constrained square-root cubature Kalman filter as a quantifying method for epileptic seizures in a mouse model. This is a useful study but the methods & results need revisions as mentioned below:

1- Methods: age of the mice is mandatory

2- Mention weight as mean+-SD & why the range is very broad? this is not prefered!! and needs justification.

3- Ethical approval number is mandatory

4- Authors should give the source of chemicals, kits and antibodies completely and consistently (code, company, town, state and country) & version for software. E.g Kainate, isoflurane & others 

5- Use appropriate abbreviations, e.g h for hours...etc

6- No statistical analysis is shown in Figure 3, Figure 4 & Figure 5 ?!!!

7- Fig 3,4&5&6 : needs to be enhanced and improved to be more clear, enlarge the panels and redistribute them.

8- In each figure leegnd: mention how data are presented & n : e.g mean+-Sd or medians in box plots...etc

Also mention this in Methods

9 -Do authors find Figure 8 best place in Discussion?

10- Authors need to compare in discussion their new method to standard methods like Racine scale to explore the value of their results.

Author Response

Response to Reviewers

Dear Reviewers

We highly appreciate your valuable comments. The manuscript has been carefully revised accordingly. Notably, the revised and added contexts of the manuscript are marked in Blue. The point-to-point responses are shown below.

          Because all 3 reviewers raised questions related to the issues around “severity”, we synthesize a reply here, before point to point reply to each reviewer.

          Sorry for the whole confusion raised by our inappropriate use of the expression “seizure severity” and “non-existence of a gold standard of seizure severity”. We understand that seizure severity is very vague. We also found that seizure severity is more used clinically to describe the attack frequency and extent. Actually, our main interest is to estimate the excitability of local neural network from the recorded EEG. We assumed that the severity of epileptiform discharge was correlated with the excitability of the local neural network, i.e., higher excitability meant more severe and we only focused on EEG waveforms. We will change the term “seizure severity” to “network excitability”.

          What our goal in long term is to formulate a quantitative indicator from EEG that correlates with the excitability of the local network, so that an automatic controller can be designed to control the excitability by electrical stimulation and therefore to suppress seizures. So the quantified severity in this manuscript implied the excitability of the local neural network and was represented by the ratio of inhibitory weight to excitatory weight of the neural network.

          We recognize and respect the value of expert opinions. Expert opinion and judgement is the final golden standard in the process of our development of quantifying seizure severity and automatic seizure suppression system.

          Racine scale was developed for the behavioral scoring of seizure severity of PTZ-induced seizures in mice. The relationship between the excitability of the network to the EEG waveform, then to the behavioral manifestation is an interesting and important topic but can be very complicated and beyond the scope of this manuscript.

          In short, in this revision,

  • “Seizure severity” will be changed to “excitability of the neural network” and the associated “severity of epileptiform discharge”. We will focus more on the lack of supportive evidences in quantitative indicators, instead of the lack of golden standard.
  • The last part of Abstract is revised; in Introduction, “quantified severity” is more clearly defined as the excitability of the local neural network and the effects and mechanisms of electrical stimulation are provided; in Methods, information about animals, drugs and devices are added; in Results, Figure legends are expanded and the bifurcation analysis is moved from Discussion to Results here; and finally, in Discussion, discussion with the published results are described.

Responses to Reviewer 1

Comments and Suggestions for Authors

Paper titled (Using constrained square-root cubature Kalman filter for quantifying the severity of epileptic activities in mice) by Huang et al. studied the utility of constrained square-root cubature Kalman filter as a quantifying method for epileptic seizures in a mouse model. This is a useful study but the methods & results need revisions as mentioned below:

1- Methods: age of the mice is mandatory

Reply:  Age (46.2±5.6 weeks) of the mice was added as suggested.

(Modification at Line 126 in the manuscript)

2- Mention weight as mean+-SD & why the range is very broad? this is not preferred!! and needs justification.

Reply:  Body weight (33.3±4.5 g) was added as suggested. The range (20 to 40 g) was the limits written in the original proposal. Actually the BW of mice used in the experiments were between 26 to 38 g.

(Modification at Line 126 in the manuscript)

3- Ethical approval number is mandatory.

Reply:  The details of Ethical approval were stated at the end of the manuscript, according to the standard template provided by the journal. The sentence is copied to the Materials section as suggested by the reviewer.

(Modification at Lines 89-91 in the manuscript)

4- Authors should give the source of chemicals, kits and antibodies completely and consistently (code, company, town, state and country) & version for software. E.g Kainate, isoflurane & others

Reply:  The information is supplied and corrected as suggested.

(Modification at Lines 94, 100-101 and 103-104 in the manuscript)

5- Use appropriate abbreviations, e.g h for hours...etc

Reply:  Modified as suggested.  

6- No statistical analysis is shown in Figure 3, Figure 4 & Figure 5 ?!!!

Reply:  Statistical analysis is added to each Figure, as suggested.

7- Fig 3,4&5&6 : needs to be enhanced and improved to be more clear, enlarge the panels and redistribute them.

Reply:  All 4 Figures are enlarged for clarity.

8- In each figure legend: mention how data are presented & n : e.g mean+-Sd or medians in box plots...etc

Reply:  Legends are expanded as suggested.

(Modification at Lines 277, 306, 338 and 363 in the manuscript)

Also mention this in Methods

Reply:  Add as suggested.

(Modification at Lines 234-240 in the manuscript)

9 -Do authors find Figure 8 best place in Discussion?

Reply:  Figure 8 and the related paragraphs are moved to the Results section.

(Modification at Lines 375-415 in the manuscript)

10- Authors need to compare in discussion their new method to standard methods like Racine scale to explore the value of their results.

Reply:  Racine scale was developed for the behavioral scoring of seizure severity of PTZ-induced seizures in mice. We will change the word “seizure severity” to “severity of epileptiform discharge” or “excitability of local neural network”, where it is appropriate.

*Please also see the synthesized explanation at the foremost of this reply.

Reviewer 2 Report

The paper uses and compares different methods to quantify parameters of the spontaneous and simulated electrical activity of the hippocampus, in a model that uses different electrical stimulation protocols.

There is, however, one basic aspect that, first of all, needs to be defined. And that is what is meant by "seizure severity". Nor is it clear whether the activity studied is clinical critical, subclinical critical or intercritical epileptiform; the differences between epileptiform and critical are not only quantitative and in particular the characteristic anisomorphism of the evolution of critical discharges involves very different patterns at the beginning of the discharge and in its overt evolution.

It is necessary that these points are preliminarily clarified. And it is also appropriate that a brief description of the supposed mechanisms of action of the various types of electrical stimulation used is provided in the introduction.

Author Response

Response to Reviewers

Dear Reviewers

We highly appreciate your valuable comments. The manuscript has been carefully revised accordingly. Notably, the revised and added contexts of the manuscript are marked in Blue. The point-to-point responses are shown below.

          Because all 3 reviewers raised questions related to the issues around “severity”, we synthesize a reply here, before point to point reply to each reviewer.

          Sorry for the whole confusion raised by our inappropriate use of the expression “seizure severity” and “non-existence of a gold standard of seizure severity”. We understand that seizure severity is very vague. We also found that seizure severity is more used clinically to describe the attack frequency and extent. Actually, our main interest is to estimate the excitability of local neural network from the recorded EEG. We assumed that the severity of epileptiform discharge was correlated with the excitability of the local neural network, i.e., higher excitability meant more severe and we only focused on EEG waveforms. We will change the term “seizure severity” to “network excitability”.

          What our goal in long term is to formulate a quantitative indicator from EEG that correlates with the excitability of the local network, so that an automatic controller can be designed to control the excitability by electrical stimulation and therefore to suppress seizures. So the quantified severity in this manuscript implied the excitability of the local neural network and was represented by the ratio of inhibitory weight to excitatory weight of the neural network.

          We recognize and respect the value of expert opinions. Expert opinion and judgement is the final golden standard in the process of our development of quantifying seizure severity and automatic seizure suppression system.

          Racine scale was developed for the behavioral scoring of seizure severity of PTZ-induced seizures in mice. The relationship between the excitability of the network to the EEG waveform, then to the behavioral manifestation is an interesting and important topic but can be very complicated and beyond the scope of this manuscript.

          In short, in this revision,

  • “Seizure severity” will be changed to “excitability of the neural network” and the associated “severity of epileptiform discharge”. We will focus more on the lack of supportive evidences in quantitative indicators, instead of the lack of golden standard.
  • The last part of Abstract is revised; in Introduction, “quantified severity” is more clearly defined as the excitability of the local neural network and the effects and mechanisms of electrical stimulation are provided; in Methods, information about animals, drugs and devices are added; in Results, Figure legends are expanded and the bifurcation analysis is moved from Discussion to Results here; and finally, in Discussion, discussion with the published results are described.

Responses to Reviewer 2

The paper uses and compares different methods to quantify parameters of the spontaneous and simulated electrical activity of the hippocampus, in a model that uses different electrical stimulation protocols.

There is, however, one basic aspect that, first of all, needs to be defined. And that is what is meant by "seizure severity". Nor is it clear whether the activity studied is clinical critical, subclinical critical or intercritical epileptiform; the differences between epileptiform and critical are not only quantitative and in particular the characteristic anisomorphism of the evolution of critical discharges involves very different patterns at the beginning of the discharge and in its overt evolution.

Reply:  We agree with the reviewer’s point. “Severity” needs to be more clearly defined. We find our usage of seizure severity is very vague. In literature, seizure severity is more used clinically to describe the attack frequency and extent. However, in this paper, we focused on EEG in the acute status epilepticus induced by Kainate. We postulated that the severity was correlated with the excitability, i.e., higher excitability meant more severe and we only focused on EEG manifestation.

The relationship between the excitability of the network to the EEG waveform, then to the behavioral manifestation is an interesting and important topic but can be very complicated and beyond the scope of this manuscript.

We will change the word “seizure severity” to “severity of epileptiform discharge” or “excitability of local neural network”, where it is appropriate.

(Please also see the synthesized explanation at the foremost of this reply.)

It is necessary that these points are preliminarily clarified. And it is also appropriate that a brief description of the supposed mechanisms of action of the various types of electrical stimulation used is provided in the introduction.

Reply:  As suggested, a brief description about the possible mechanisms of action of the two types of electrical stimulation used is provided in Introduction.

(Modification at Lines 41-54 in the manuscript)

Reviewer 3 Report

"Using constrained square-root cubature Kalman filter for quantifying the severity of epileptic activities in mice" is an interesting article. The main purpose of this study was to investigate the eligibility of using constrained square-root cubature Kalman filter (CSCKF) to quantify the severity of epileptic activities. The performance of CSCKF was compared with three commonly used indicators: band power, Teager energy operator and sample entropy, using epileptic activities from both experiments and a computational seizure model.

There are some issues in the article that need to be addressed as discussed below.

Abstract

Lines 25 and 26 mention   Conclusion: and in the next line In conclusion… the wording is confusing and redundant, it is suggested to rewrite this part of the  manuscript. 

Introduction

In line 49, a crucial and very important part is how to evaluate the seizure severity and it is mentioned that one of them is through the detection of spikes in the EEG. However, the article only superficially describes a single work on this topic. By the way, the article refers to tumors and epileptic activity and not to the severity of seizures in temporal lobe epilepsy, which is the experimental model used in the article. It is suggested to provide more information on this aspect and support it with the corresponding references.

Discussion

In line 339 it is mentioned that the results when comparing the four proposed techniques did not show differences between them. And furthermore, there is no gold standard to determine seizure severity, although expert opinion has been used.  

It is suggested:

1.- Add the discussion of the non-existence of a gold standard supported by the corresponding references.

2.- Add and discuss the references that support that expert opinion is not a gold standard to determine seizure severity

3.- Justify why expert opinion was not used as the gold standard.

In the entire discussion there is only one reference, on the contrary it is intended to discuss and base on a series of additional experiments that try to explain why there are no differences between the four proposed techniques. It is suggested to place this series of additional experiments in the results section and in the discussion section dedicate it to the comparison with the results obtained in other studies with the corresponding references.

In line 350 Figure 1A is mentioned, it is an error and they refer to A1 ?

References

There are some journal names that start with capital letters and others with lower case. Check carefully and correct them.  

Author Response

Response to Reviewers

Dear Reviewers

We highly appreciate your valuable comments. The manuscript has been carefully revised accordingly. Notably, the revised and added contexts of the manuscript are marked in Blue. The point-to-point responses are shown below.

          Because all 3 reviewers raised questions related to the issues around “severity”, we synthesize a reply here, before point to point reply to each reviewer.

          Sorry for the whole confusion raised by our inappropriate use of the expression “seizure severity” and “non-existence of a gold standard of seizure severity”. We understand that seizure severity is very vague. We also found that seizure severity is more used clinically to describe the attack frequency and extent. Actually, our main interest is to estimate the excitability of local neural network from the recorded EEG. We assumed that the severity of epileptiform discharge was correlated with the excitability of the local neural network, i.e., higher excitability meant more severe and we only focused on EEG waveforms. We will change the term “seizure severity” to “network excitability”.

          What our goal in long term is to formulate a quantitative indicator from EEG that correlates with the excitability of the local network, so that an automatic controller can be designed to control the excitability by electrical stimulation and therefore to suppress seizures. So the quantified severity in this manuscript implied the excitability of the local neural network and was represented by the ratio of inhibitory weight to excitatory weight of the neural network.

          We recognize and respect the value of expert opinions. Expert opinion and judgement is the final golden standard in the process of our development of quantifying seizure severity and automatic seizure suppression system.

          Racine scale was developed for the behavioral scoring of seizure severity of PTZ-induced seizures in mice. The relationship between the excitability of the network to the EEG waveform, then to the behavioral manifestation is an interesting and important topic but can be very complicated and beyond the scope of this manuscript.

          In short, in this revision,

  • “Seizure severity” will be changed to “excitability of the neural network” and the associated “severity of epileptiform discharge”. We will focus more on the lack of supportive evidences in quantitative indicators, instead of the lack of golden standard.
  • The last part of Abstract is revised; in Introduction, “quantified severity” is more clearly defined as the excitability of the local neural network and the effects and mechanisms of electrical stimulation are provided; in Methods, information about animals, drugs and devices are added; in Results, Figure legends are expanded and the bifurcation analysis is moved from Discussion to Results here; and finally, in Discussion, discussion with the published results are described.

Responses to Reviewer 3

"Using constrained square-root cubature Kalman filter for quantifying the severity of epileptic activities in mice" is an interesting article. The main purpose of this study was to investigate the eligibility of using constrained square-root cubature Kalman filter (CSCKF) to quantify the severity of epileptic activities. The performance of CSCKF was compared with three commonly used indicators: band power, Teager energy operator and sample entropy, using epileptic activities from both experiments and a computational seizure model.

There are some issues in the article that need to be addressed as discussed below.

Abstract

Lines 25 and 26 mention   Conclusion: and in the next line In conclusion… the wording is confusing and redundant, it is suggested to rewrite this part of the manuscript.

Reply:  The sentences are re-written to avoid redundancy.

(Modification at Lines 25-27 in the manuscript)

Introduction

In line 49, a crucial and very important part is how to evaluate the seizure severity and it is mentioned that one of them is through the detection of spikes in the EEG. However, the article only superficially describes a single work on this topic. By the way, the article refers to tumors and epileptic activity and not to the severity of seizures in temporal lobe epilepsy, which is the experimental model used in the article. It is suggested to provide more information on this aspect and support it with the corresponding references.

Reply:  Sorry for the confusion by our vague use of seizure severity. Our goal is to quantify the severity of epileptiform discharge due to different levels of neural excitability. The description in Introduction about the quantification of severity is expanded.

(Modification at Lines 65-71 in the manuscript)

Discussion

In line 339 it is mentioned that the results when comparing the four proposed techniques did not show differences between them. And furthermore, there is no gold standard to determine seizure severity, although expert opinion has been used. 

It is suggested:

1.- Add the discussion of the non-existence of a gold standard supported by the corresponding references.

2.- Add and discuss the references that support that expert opinion is not a gold standard to determine seizure severity

3.- Justify why expert opinion was not used as the gold standard.

Reply:  Sorry for the whole confusion raised by our inappropriate use of the expression “non-existence of a gold standard of seizure severity”. As explained in the above (reply to the second reviewer), severity is a complex term and needs to be defined more clearly. What we try to do is to formulate a quantitative indicator from EEG that correlates with the excitability of the local network, so that an automatic controller can be designed to control the excitability by electrical stimulation and therefore the seizure.

Of course, the final golden standard is the expert opinion. Any quantitative indicator needs to match the opinion and judgements of experts. We have not come to this step yet. This paper only found that the DC offset of EEG is important in quantifying the “severity” or the excitability of the local neural network. We came to this conclusion from simulation study. In the next step, we need to do experiments using apparatus that can record DC component of EEG to verify our finding of this manuscript. Expert opinion and judgement is the final golden standard in this process of development.

The descriptions in the manuscript are modified to reflect the above explanation.

In the entire discussion there is only one reference, on the contrary it is intended to discuss and base on a series of additional experiments that try to explain why there are no differences between the four proposed techniques. It is suggested to place this series of additional experiments in the results section and in the discussion section dedicate it to the comparison with the results obtained in other studies with the corresponding references.

Reply:  As suggested, Figure 8 and the whole paragraph are moved to the Results section.

In discussion, we compared the performance of the 4 indicators because these 4 indicators were what we summarized from literature. Most studies used the spike frequency or spike amplitude as the indicator. Nonlinear indicator, such as entropy, was developed in more recent years. The discussion is expanded to cover the comparison with results of more studies.

(Modification at Lines 432-439 and 454-473 in the manuscript)

In line 350 Figure 1A is mentioned, it is an error and they refer to A1 ?

Reply:  The typo is corrected.

(Modification at Line 451 in the manuscript)

References

There are some journal names that start with capital letters and others with lower case. Check carefully and correct them. 

Reply:  Thanks for the careful checking. In fact, we used the file template supplied by the journal and used EndNote for formatting references. We will re-check the format after final update.

(Modification at Line 519 References in the manuscript)

Round 2

Reviewer 1 Report

Thanks

Author Response

Thanks for the encouragement. 

Reviewer 2 Report

The changes and additions made to the current version of the manuscript have the advantage of better explaining which is the object of study of the work, namely the "network excitability".

However, it seems to me that the use of sophisticated methods does not make a significant contribution in the quantification of a parameter that presents, first of all, a problematic definition also from a qualitative point of view. Not only that, but if the purpose of the study is to evaluate the different power of performance indicators, a "competition ground" whose characteristics are not precisely known does not seem suitable. It is therefore better to insist on simulated data (the characteristics of which are known, being established by the experimenters) before applying the performance indicators to experimental epilepsy in animal models.
